# Artificial Intelligence for Early Detection of Chest Nodules in X-ray Images

**DOI:** 10.3390/biomedicines10112839

**Published:** 2022-11-07

**Authors:** Hwa-Yen Chiu, Rita Huan-Ting Peng, Yi-Chian Lin, Ting-Wei Wang, Ya-Xuan Yang, Ying-Ying Chen, Mei-Han Wu, Tsu-Hui Shiao, Heng-Sheng Chao, Yuh-Min Chen, Yu-Te Wu

**Affiliations:** 1Department of Chest Medicine, Taipei Veterans General Hospital, Taipei 112, Taiwan; 2Institute of Biophotonics, National Yang Ming Chiao Tung University, Taipei 112, Taiwan; 3Division of Internal Medicine, Hsinchu Branch, Taipei Veterans General Hospital, Hsinchu 310, Taiwan; 4School of Medicine, National Yang Ming Chiao Tung University, Taipei 112, Taiwan; 5Department of Critical Care Medicine, Taiwan Adventist Hospital, Taipei 105, Taiwan; 6Department of Medical Imaging, Cheng Hsin General Hospital, Taipei 112, Taiwan; 7Department of Radiology, Taipei Veterans General Hospital, Taipei 112, Taiwan; 8Institute of Biomedical Informatics, National Yang Ming Chiao Tung University, Taipei 112, Taiwan; 9Brain Research Center, National Yang Ming Chiao Tung University, Taipei 112, Taiwan

**Keywords:** artificial intelligence, AI, detection, lung cancer, machine learning

## Abstract

Early detection increases overall survival among patients with lung cancer. This study formulated a machine learning method that processes chest X-rays (CXRs) to detect lung cancer early. After we preprocessed our dataset using monochrome and brightness correction, we used different kinds of preprocessing methods to enhance image contrast and then used U-net to perform lung segmentation. We used 559 CXRs with a single lung nodule labeled by experts to train a You Only Look Once version 4 (YOLOv4) deep-learning architecture to detect lung nodules. In a testing dataset of 100 CXRs from patients at Taipei Veterans General Hospital and 154 CXRs from the Japanese Society of Radiological Technology dataset, the sensitivity of the AI model using a combination of different preprocessing methods performed the best at 79%, with 3.04 false positives per image. We then tested the AI by using 383 sets of CXRs obtained in the past 5 years prior to lung cancer diagnoses. The median time from detection to diagnosis for radiologists assisted with AI was 46 (3–523) days, longer than that for radiologists (8 (0–263) days). The AI model can assist radiologists in the early detection of lung nodules.

## 1. Introduction

Lung cancer is the leading cause of death in patients with neoplasm in Taiwan. Patients with lung cancer are usually asymptomatic or have nonspecific complaints; however, more than half of initial lung cancer diagnoses are stage IIIB or higher, meaning the tumors are unresectable [1]. One method of improving lung cancer survival rate is early screening. Both the National Lung Screening Trial (NLST) and the Dutch–Belgian Randomized Lung Cancer Screening (NELSON) trial revealed that early detection of lung cancer resulted in a 20% improvement in overall survival [2,3]. Therefore, effective screening tools for early diagnosis of lung cancer warrant investigation.

The standard procedures for lung cancer screening are chest X-rays (CXRs) and low-dose chest computed tomography (LDCT). LDCT is currently the most powerful diagnostic tool, with a resolution as fine as 1 mm, whereas nodules must be larger than 1 cm to be detectable on a CXR [4]. Several studies have investigated nodule detection in chest computed tomography (CT) images and found that artificial intelligence (AI) outperforms humans in this field [5]. The incidence of lung cancer was approximately 6 cases per 1000 person-years in both the NLST and the NELSON trials [2,3]. Active LDCT screening is only recommended for high-risk groups [6]. Screening with CXR is more cost-effective and has greater availability; hence, hospitals have used it to increase lung cancer screening coverage. However, the initial diagnosis rate of nodules using CXRs is relatively low. The sensitivity is approximately 76%, even for expert screening [7]. One study demonstrated that as many as 90% of missed lung cancer cases were detected through retrospective review [8]. AI is an accurate and stable method of detecting lung nodules and may be useful as a second opinion.

Advances in computing power and the application of such power to large datasets have led to the rapid growth in AI image analysis. Several algorithms, such as the You Only Look Once (YOLO) algorithm, and convolutional neural networks have been used in image detection and classification [9,10]. Computer-aided diagnosis has assisted radiologists when reading CXRs for lung nodule detection [11]. However, most research has focused on the AI structure and evaluating AI performance by detection rate but has ignored the influence of image preprocessing and the performance of AI models on a temporal scale. Therefore, we constructed an AI nodule detector and then evaluated it in a retrospective analysis, where different preprocessing techniques were applied.

## 2. Materials and Methods

### 2.1. Dataset

We retrospectively enrolled patients diagnosed with single small lung nodules at any time between 2011 and 2014 in Taipei Veterans General Hospital (VGHTPE), a tertiary medical center in northern Taiwan. We enrolled patients who had undergone biopsy or surgical procedures. The inclusion criteria were as follows: (1) age 20 or over; (2) received CT-guided biopsy or wedge resection; and (3) newly diagnosed lung nodule on chest CT. The exclusion criterion was patients with multiple lung nodules on chest CT scans. Patients’ initial chest CT scans and CXRs before diagnosis were collected as the VGHTPE dataset (Table 1). To ensure accuracy, one senior radiologist and two pulmonology fellows annotated the nodules in the CXRs and compared the annotations with the chest CT scans. We held weekly meetings to discuss unclear cases, and final decisions were made by voting. The annotated CXRs were separated into the VGHTPE training dataset for training and validation, and a testing dataset for testing. This study was approved by the Institutional Review Board of Taipei Veterans General Hospital (VGHIRB No.2018-04-11AC).

We used four open datasets (Table 1) for training and testing. The first was the Shenzhen dataset, collected in collaboration with Shenzhen No. 3 People’s Hospital, Guangdong Medical College, Shenzhen, China. The Shenzhen dataset contained segmented lung masks manually delineated by Stirenko and colleagues [12]. The second was the Montgomery dataset, which contained manually segmented lung masks and was collected in collaboration with the Department of Health and Human Services, Montgomery County, MD, USA [13]. The third was the National Institutes of Health (NIH) chest X-ray dataset [14], which contained 112,120 X-ray images with labels. The fourth was a standard digital image database created by the Japanese Society of Radiological Technology (JSRT) in cooperation with the Japanese Radiological Society [15].
biomedicines-10-02839-t001_Table 1Table 1Introduction of datasets.DatasetNumber of CXRsDescriptionApplicationVGHTPE training559659 CXRs with biopsy confirmed lung nodules obtained from VGHTPE and separated into 559 CXRs for training dataset and 100 CXRs isolated for testing dataset.TrainingVGHTPE testing100TestingShenzhen [13]662662 CXRs with segmented lung masks manually delineated by Stirenko and colleagues.SegmentationMontgomery [13]13880 normal CXRs and 58 CXRs with tuberculosis manifestations. All with lung masks.SegmentationNIH [14]880880 CXRs with annotations in 112,120 CXRs from NIH open datasetPretrainJSRT [15]154154 CXRs with lung nodules in 247 CXRs collected by JSRT.TestingJSRT—the Japanese Society of Radiological Technology; NIH—National Institutes of Health; VGHTPE—Taipei Veterans General Hospital.

### 2.2. Machine Learning Pipeline

Our AI nodule detector pipeline comprised three steps. First, all CXRs underwent monochrome preprocessing and brightness correction to normalize the intensities from different CXR machines. The CXRs were then processed into three separate datasets to enhance computer vision. Second, a bounding box was created to cover the segmented lungs, mediastinum, and subdiaphragmatic region using the Shenzhen dataset for U-net training and the Montgomery dataset for segmenting the lung areas [13]. Finally, we trained the AI model based on the YOLOv4 algorithm [10], which was pretrained on the NIH chest X-ray dataset [14], to detect the location of lung nodules and validate the findings by cross-validating with the VGHTPE training dataset. We then tested the AI model using the VGHTPE testing dataset and the JSRT dataset to evaluate its performance (Figure 1).

### 2.3. Preprocessing

CXR quality varied between different X-ray machines and radiology technicians. Therefore, we preprocessed the CXRs primarily to normalize the CXRs to judge images by the same standard. In this study, we applied a preprocessing step (Preprocessing I: monochrome and brightness correction) that was applied to all images. The details of this process are described in Section 2.3.1. Furthermore, to enhance the images, different preprocessing methods (Preprocessing II), such as gamma correction (Section 2.3.2) and contrast limited adaptive histogram equalization (CLAHE; Section 2.3.3), were applied to enhance the contrast and to decrease noise. Three identical datasets were produced using three different Preprocessing II methods. The whole process of the preprocessing procedure is illustrated in Figure 2.

Three identical datasets were produced from the original VGHTPE training dataset and VGHTPE testing dataset using Preprocessing I and the different Preprocessing II methods.

#### 2.3.1. Image Normalization

The image normalization included two steps: monochrome correction and brightness correction. The default settings of the X-ray machines varied. During our manual screening, the grayscale of some images were revealed to be inverse (Figure 3). We corrected it according to the four edges of the image to maintain grayscale consistency.

The brightness of the CXRs varied with different exposure settings. We removed the top 0.2% of the image intensity from the graph and rescaled the histogram to normalize the contrast of the CXRs (Figure 4).

#### 2.3.2. Gamma Correction

After default normalization, we utilized the gamma correction method to increase the contrast of the images [16], in which the pixels of images could be manipulated with the following equation:(1)$$V_{out}=AV_{in}^{\gamma }$$
where $V_{in}$ is the input value of each pixels, and $V_{out}$ indicates the output value of each pixel. *A* is a constant (usually, *A* = 1), and γ is the gamma value, in the form of the power-law. When γ < 1, the dark areas were lighter, and the bright areas were darker, which entailed a more diverse intensity distribution. As the midlevel decreased, the intensity distribution became more diverse and more skewed toward extreme values. Moreover, the shadows became darker when  γ > 1, the contrast of midlevel values was enhanced, and the distribution of the extreme values was compressed. This method can focus on high or low contrast zones with different gamma values. High intensity with lower contrast zones, such as soft tissue and bone, could be emphasized with γ < 1, whereas high contrast zones like lung area could be emphasized by γ > 1.

In this study, we set γ = 3 to demonstrate the midlevel grayscale area of the images and sharpen the borders of high contrast zones.

#### 2.3.3. CLAHE

Another technique used to improve the contrast in this study was CLAHE, a variant of adaptive histogram equalization (AHE). It corrected the images based on the distribution of the histograms by dividing images into the blocks we set and by applying histogram equalization to each block. This method redistributes the lightness value of small pieces of the image to improve the local contrast but prevents overamplifying noise in relatively homogeneous areas, in a similar manner to AHE [17]. Therefore, two main parameters should be determined when applying CLAHE: the clip limit (the threshold for contrast limiting) and the grid size (number of tiles in the row and column). In this study, we set the clip limit to 2 and the grid size to 8 × 8.

### 2.4. AI Models

We used two AI models: U-net for lung segmentation and YOLOv4 for lung nodule detection. Three models were produced according to three derived datasets that underwent default preprocessing, gamma correction, or CLAHE. The fourth model was then derived from the three previous models by union method after the detection threshold was adjusted (Figure 5).

#### 2.4.1. U-net

U-net is a convolutional network architecture used for precise and fast image segmentation and can be trained end to end with few images. In the 2015 International Symposium on the Biomedical Imaging Cell-Tracking Challenge, U-net outperformed the previous segmentation method, a sliding-window convolutional network, by a large margin [18]. The open Shenzhen dataset and Montgomery datasets were used to train the U-net for lung segmentation. The parameters in our study were an image size of 512 × 512 pixels, an epoch of 56, a batch size of 302, and a learning rate of 0.00001. The validation accuracy was approximately 0.93. We constructed the bounding box area with algorithms in Open Source Computer Vision Library (OpenCV) [19] to cover the segmented lung, the retro-cardiac region, and the subdiaphragmatic area.

#### 2.4.2. YOLOv4

YOLOv4 is the fourth iteration of the YOLO algorithm [20] and is better than YOLOv3 in terms of speed, detection accuracy, and hardware requirements. YOLOv4 achieved 43.5% average precision (AP) and 65.7% AP at an intersection over union of 0.5 (AP50) on the Microsoft Common Objects in Context (MSCOCO) dataset [21], a large dataset containing common objects in our everyday life. YOLOv4 is 10% more accurate and 12% faster than is YOLOv3. We pretrained YOLOv4 with 880 annotated CXRs from the NIH dataset before training the model for nodule detection. The NIH dataset contained 112,120 CXRs with >90% accurately labeled lesions. We then trained YOLOv4 with the preprocessed VGHTPE training dataset and further derived datasets, including the Gamma correction-VGHTPE training dataset and the CLAHE-VGHTPE dataset shown in Figure 5. The parameters in our study were nine anchor boxes produced by k-means, an image size of 512 × 512 pixels, an epoch of 500 (early stopping by the value of loss function at a patience of 50), a batch size of 3, and a learning rate of 0.001 (reduced by value of loss function with a factor of 0.5 and a patience of 20).

### 2.5. Performance Evaluation

The performance of the four lung nodule detection models were presented with free-response receiver operating characteristic (FROC) curves. We compared the area under curve of the FROC curves of the four models to choose the best model. We applied the final lung nodule detection model to the VGHTPE testing dataset for testing and the JSRT dataset for external validation (Figure 6).

### 2.6. Temporal Analysis

CXRs obtained in the past 5 years before the diagnosis of early-stage lung cancer between 2011 and 2014 were also collected from the VGHTPE for the retrospective review. Pulmonologists manually reviewed the CXR reports. We considered the nodules “detected by radiologists” if the radiology reports identified nodules in the correct lung region. The trained YOLOv4 was also used for detection. The nodules were labeled “detected by the AI model” if the location detected by YOLOv4 was correct about the CXR taken at diagnosis. We then combined the radiology reports and the results from our AI model through the set union operation to simulate how radiologists would perform with AI assistance. The performances of the radiologists, the AI model, and radiologists assisted by the AI model were then compared (Figure 7).

### 2.7. Statistical Analysis

SPSS (version 22.0, SPSS Inc., Chicago, IL, USA) was used for the statistical analysis program. Continuous variables were examined for normality with the Kolmogorov–Smirnov test. Variables are presented as means ± standard deviations if normally distributed and as the median (interquartile range) if nonnormally distributed. A waterfall plot was used to model the effect of machine learning on early detection. A Wilcoxon signed-rank test was used to compare the performance (time from detection to diagnosis) of the radiologists, the AI model, and radiologists assisted by the AI model.

### 2.8. Ethics

This study was approved by the VGHIRB. All procedures were conducted per the ethical principles of the VGHIRB, who approved the study protocol (approval number 2018–10-003CC). The VGHIRB did not require parental or participant consent for reviewing patient medical records due to the retrospective study design. Patient data confidentiality was ensured through compliance with the Declaration of Helsinki.

## 3. Results

We obtained 659 pairs of CXRs and chest CT scans from patients with newly biopsied or resected lung nodules between 2011 and 2014. Table 2 presents the characteristics of both groups. The median age of the patients with CXRs displaying lung nodules was 65 years of age (56–72 years of age). A total of 34 (5.2%), 44 (6.7%), 260 (39.5%), 163 (24.7%), and 158 (24.0%) nodules were <6 mm, 6 mm–1 cm, 1–2 cm, 2–3 cm, and >3 cm, respectively.

Out of the 659 CXRs with lung nodules, 559 were used for training and validation, and 100 for testing. The CXRs were normalized and preprocessed into three datasets with default normalization, gamma correction, and CLAHE, respectively. Three AI models were separately trained with the three preprocessed VGHTPE training datasets and then merged into the ensemble model.

### 3.1. Comparison of Preprocessing

We constructed the lung nodule prediction model from the 559 CXRs of the VGHTPE training dataset. The CXRs were sent for model training after preprocessing with default normalization, gamma correction, and CLAHE method. We built the original model (Figure 8A,D), gamma correction model (Figure 8B,E), CLAHE model (Figure 8C,F), and ensemble model (Figure 8G) to predict the location of lung nodules based on YOLOv4 structures. Figure 8 displays the preprocessed images and red bounding boxes formed with the YOLOv4-based models. The location of the nodule is marked by a yellow circle.

The performance of the four models was tested using the 100 CXRs of the VGHTPE testing dataset. Thresholds of 0.1, 0.075, 0.05, 0.02, 0.01, 0.0075, 0.005, 0.0025, 0.001, and 0.0001 were used for the four models to generate the FROC curves (Figure 9). The ensemble model achieved better sensitivity (85% sensitivity at 2.8 false positives per image) with the same false positives per image and had the largest area under curve.

### 3.2. External Validation with JSRT

We performed an external validation of the ensemble model (Section 3.1) on the JSRT dataset. The basic characteristics of the VGHTPE testing dataset and the JSRT dataset are listed in Table 3.

The sensitivity of the ensemble model was 89.6% with 7.99 FPs per image when ap-plied to the VGHTPE testing dataset, and 84.4% with 8.6 FPs per image when applied to the JSRT dataset. The sensitivity increased with lung nodule size (Figure 10).

### 3.3. Temporal Analysis

A total of 383 sets of CXRs with malignant nodules were collected. The characteristics of the CXR at diagnosis are listed in Table 4. All of the nodules on the CXRs at the time of lung cancer diagnosis were larger than 6 mm.

We then compared the performance of the radiologists with that of the AI model (ensemble model). Among the 383 patients with lung cancer, 226 patients had nodules on CXRs detected early by the AI model, while radiologists discovered nodules early in 203 patients. Comparing the detections of AI and radiologists, the nodules were detected early in the same 135 patients through both the AI model and radiologists. The AI model detected 91 other cases that the radiologists did not. Conversely, radiologists detected only 68 more patients’ nodules that the AI did not detect early (Figure 11A). Nodules of 89 patients were not detected early by either the AI model or radiologists. Regarding the median time from detection to diagnosis, among total 383 sets of CXRs, the AI model detected nodules 19 (0–199) days early, whereas radiologists detected these only 8 (0–263) days early. However, in the estimation of the radiologists’ performance assisted with the AI model, the median time from detection to diagnosis increased to 46 (3–523) days. With consideration to the effect of nodule size, Appendix A lists the probability of early detection according to nodule size. Both the AI model (ensemble model) and the radiologists performed better when the nodule size increased (Appendix A).

For the 135 sets of CXRs that contained a single nodule detected early by both the AI model and radiologists, the median time from detection to diagnosis is presented in the central part of Figure 11A. The time was significantly longer for the radiologists assisted by the AI model (303 (43–1007) days) than for the radiologists (194 (32–889) days, *p* < 0.001) or than for the AI model (210 (39–776) days, *p* < 0.001). In addition, the median time from detection to diagnosis of the 91 patients detected by only the AI but undiscovered by radiologists was 63 (21–392) days. On the other hand, for the 68 patients discovered by only radiologists without AI assistance was 150 (28–676.5) days. Figure 11B displays the performance in the early detection of nodules on the CXRs of all 383 patients. The green line shows that the performance of the radiologists assisted with the AI model was better than that of only the AI model or radiologists. Figure 11C presents 156 sets of CXRs that benefited from the assistance of the AI model, including the 91 cases that were only detected early using AI and 65 cases that were discovered early by both AI and radiologists. The AI model and radiologists performed the same in 70 patients.

## 4. Discussion

In this retrospective case–control study, four YOLOv4-based lung nodule detection AI models were built with differently preprocessed CXRs. We compared the models with the VGHTPE testing dataset for internal validation, and the ensemble model performed the best. The ensemble model was then externally validated using the JSRT open dataset with a sensitivity of 84.4%. We tested the AI model and compared its performance with that of radiologists on a temporal scale. Although most studies have focused on differences in accuracy between AI models and doctors, we extended the scope of our investigation to the preprocessing step and temporal scale, and aimed to determine how much earlier an AI model could detect lung nodules compared to humans.

Preprocessing is key in computer vision applications. In this study, preprocessing was first used to standardize the images to overcome the intermachine or intertechnician variability. Second, the tuning of contrast or the contrast window helped the computer to focus on different structures. Third, the region of interest (ROI), referring to the lung segmentation part, might reduce computing power and reduce FPs because it excludes extrapulmonary areas. Different preprocessing steps have been used for AI models that detect COVID-19 [22,23]. An ensemble method of different preprocessing methods has been proposed for tuberculosis detection [24]. A study has investigated different ROIs affecting the predictability of nodule status according to ultrasound-based radiomic features in breast cancer patients, and it showed advantages in using the ensemble model to enhance the predictive power of the AI models [25]. However, the effect of different preprocessing on nodule detection has been less addressed. Therefore, we tested the different preprocessing steps and found that the ensemble model performed the best.

The lung segmentation technique we used was based on a conventional U-net. The performance of the technique was validated in the 2015 International Symposium on Biomedical Imaging Cell-Tracking Challenge [18]. However, we did not re-evaluate the model because many studies have tested the method [26,27,28].

The AI model performance depended on the size and the quality of the training data and the algorithm. Training datasets used for AIs are usually large. For example, Majkowska and colleagues used two large datasets containing more than 800,000 images to train an AI model to detect pneumothorax, opacity, nodules, masses, and fractures [29]. Sharma et al. used more than 4000 CXRs to train, test, and validate an AI model capable of detecting COVID-19 [30]. The sensitivity of nodule detection in previous studies has ranged from 51% to 83.3%, with FPs per image ranging from 2 to 6.4 [31]. Researchers reported a 99% sensitivity with 0.2 FPs with patch-based multiresolution convolutional networks [32]. The sensitivity of our model was 71.4% with 4.26 FPs per image and 82% with 8.4 FPs per image. We attributed to the relatively lower performance to the relatively small sample size used for training (approximately 500 CXRs) and the YOLO architecture. However, our model’s sensitivity is already comparable to those of human experts. Our AI program’s good performance may be attributable to the quality of our training dataset. The cases in our training dataset were pathologically proven, and the samples were obtained through either bronchoscopic biopsy, CT-guided biopsy, or surgical procedures; therefore, the diagnoses were credible. We also allowed doctors to simultaneously view the CT scans at the same time to ensure the nodules were labeled at the accurate locations.

Our use of a stepwise method was modeled after the human thought processes and allowed the AI model to identify the nodules. The image normalization method we used to overcome intermachine variability in the VGHTPE dataset also reduced variability between datasets. Extrapulmonary nodules and artifacts were mostly excluded from the lung segmentation. However, the retrocardiac region and subdiaphragmatic regions presented challenges during lung segmentation [33] and might have been neglected by this method. We used a U-net-trained lung segmentation model as a template to create a bounding box of the lungs covering the retrocardiac and subdiaphragmatic regions. The advancements in the YOLO algorithm increased the sensitivity of lung nodule detection and were computationally inexpensive. These factors allowed the AI model to achieve a sensitivity of approximately 85% in the external validation of the JSRT dataset, which is comparable to the sensitivity of the radiologists assisted by the AI system in other studies [11,31,34,35,36]. Another review also reported similar levels of sensitivity at 58.67% (44.2–71%) alongside a mean 2.22 (0.19–3.9) FPs per image [37]. Newer deep learning nodule detection models may have better performance [32]. However, although one of these deep learning models achieved more than 98% sensitivity on the JSRT dataset, the sensitivity dropped to below 50% in the external validation of the Guangzhou Hospital dataset and Shenzhen Hospital datasets [32]. The problem may be caused by overfitting. The trained model performed well in the training dataset and internal validation. Mixing the training, validation, and testing dataset might overestimate the performance of the AI model, as the performance dropped dramatically in external validation. In our study, our AI model provided consistent performance; therefore, the overfitting was less severe.

For the retrospective review, we used reports, rather than reinterpretations by two experts, as a reference. Consequently, the doctors’ performance might have been underestimated, and 13–60% of 1 cm nodules could have been missed in the absence of a retrospective review [8]. However, our study may reflect real-world practical settings because we used reports written by radiologists. Our results indicate that the AI model outperformed the radiologists for some CXRs, but the radiologists outperformed the AI model for others. Anna et al. reported similar results [29], suggesting that AIs may be more effective in aiding experts in making diagnoses rather than in replacing experts. Our study showed that the duration from detection to diagnosis was comparable for the AI model (210 days) and the radiologists (194 days). The duration from detection to diagnosis was longer for radiologists assisted with the AI model (303 days) among CXRs with nodules detected by both the AI and the radiologists before diagnosis. This result supports the notion that the combination of AI model and radiologists may provide benefit in the clinical scenario. However, we used the CXRs at diagnosis in training, so they might have been similar to the CXRs before diagnosis. The performance of the AI model might have been overestimated. Further external validation and additional prospective studies are still warranted.

The time from detection to diagnosis depended on many factors, such as the performance of the image diagnosis system, the growth rate and the doubling time of the tumor [38], the frequency of the CXR follow-up, and the efficiency of the health care system [39,40]. We focused on the performance of the image diagnosis system because it is a key component that first-line health care providers can improve. The diagnostic system performed better as time from detection to diagnosis increased. Our study indicated that the median time from detection to diagnosis was longer for radiologists assisted with the AI model than it was for the AI model only or radiologists in all 383 sets of CXRs (46 vs. 19 and 8 days) or the 135 sets of CXRs that were detected early by both the AI model and radiologists (303 vs. 210 and 194 days).

Several factors may limit the applicability of our study to clinical settings. The first factor is nodule size. Experts have concluded that nodules must be larger than 1 cm to be detected on CXRs [41]; however, we still included some nodules smaller than 1 cm to enrich the database. The sensitivity of nodules smaller than 1 cm decreased slightly in the interval validation on the VGHTPE testing dataset but dropped dramatically to 50% in the external validation on the JSRT dataset. Larger tumors were also excluded. Although the main objective of early screening is to detect an operable tumor, we focused on screening for stage I and stage II lung cancer. Thus, our model may be limited in detecting nodules smaller than 1 cm or masses larger than 7 cm. Second, the variation in doubling time among types for lung cancer may affect the time from detection to diagnosis. Our relatively small sample size meant that we did not divide our data according to the type of lung cancer. Third, unlike a prospective study, our study used CXRs obtained before diagnosis. Several of the CXRs had been obtained before a nodule was detected by radiologists, causing a large variance in CXR follow-up time. The variance in follow-up time might have affected the detection and the weight of the CXRs from different patients. However, this factor was the same during detection by both the AI and the radiologists. Fourth, this was a single-center study. The applicability of the AI model to an external dataset is limited despite our validating the model on the JSRT dataset. Finally, the training data did not include normal CXRs, which we frequently encounter in clinical settings. The model’s accuracy should also be validated before it is applied to nodule detection.

## 5. Conclusions

The ensemble model of different preprocessing methods can improve the performance of the AI model. Radiologists may perform better with the assistance of AI models in temporal scale. Our results indicate that our AI method can aid in the early diagnosis of lung cancer in routine clinical practice.

## Figures and Tables

**Figure 1 biomedicines-10-02839-f001:**
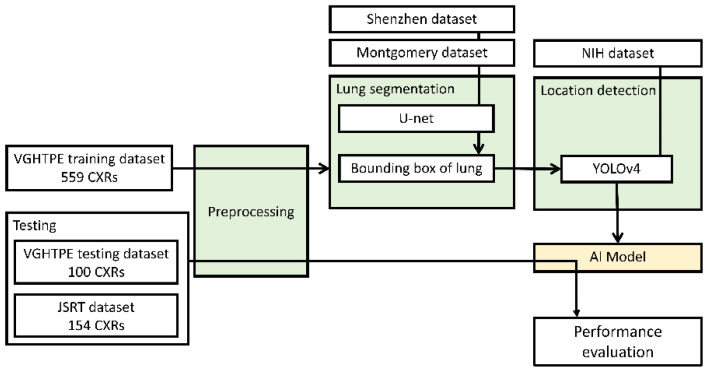
Pipeline of the study design.

**Figure 2 biomedicines-10-02839-f002:**
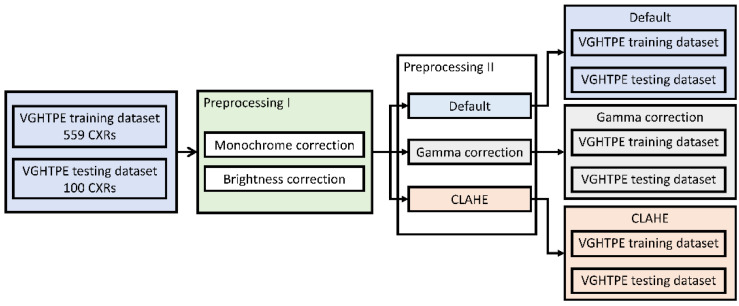
Preprocessing procedure.

**Figure 3 biomedicines-10-02839-f003:**
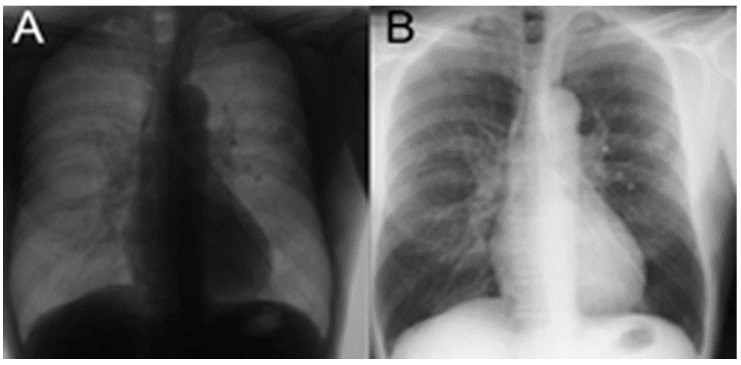
Example of the same image with inverse grayscale. (**A**) The CXR with inverse grayscale. (**B**) The corrected CXR after monochrome correction.

**Figure 4 biomedicines-10-02839-f004:**
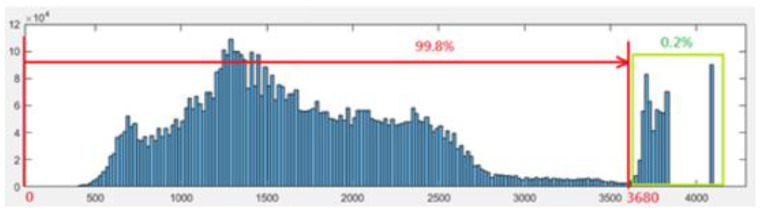
Example of removing the top 0.2% of signals from the histogram.

**Figure 5 biomedicines-10-02839-f005:**
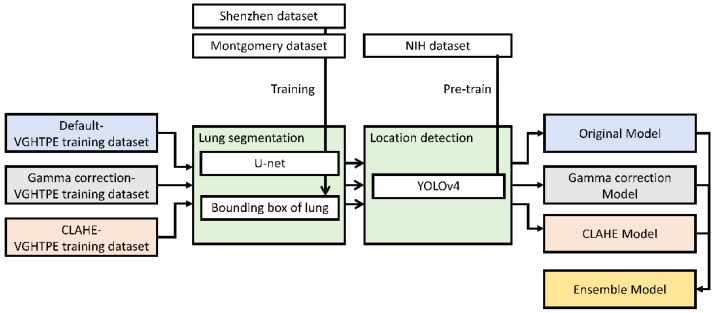
Training pipeline of the lung nodule detection models.

**Figure 6 biomedicines-10-02839-f006:**
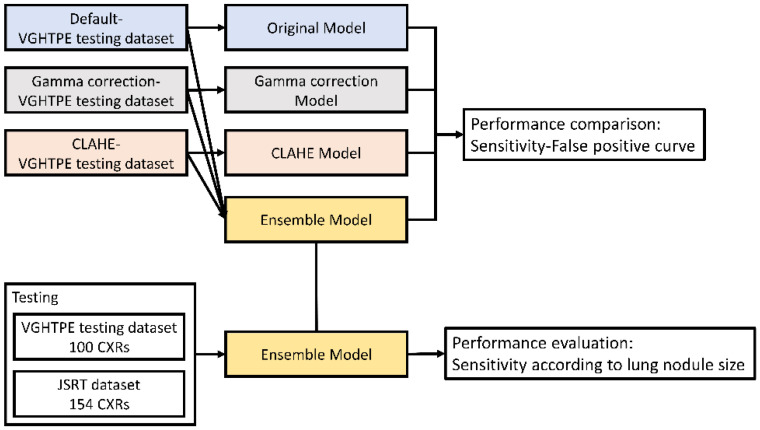
Performance evaluation of the different lung nodule detection models.

**Figure 7 biomedicines-10-02839-f007:**
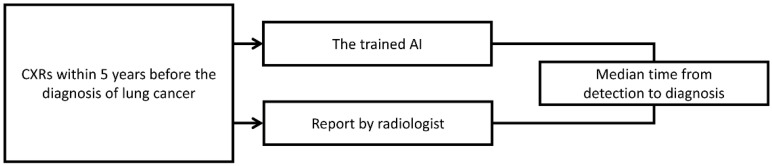
Study design of the model validation on a temporal scale.

**Figure 8 biomedicines-10-02839-f008:**
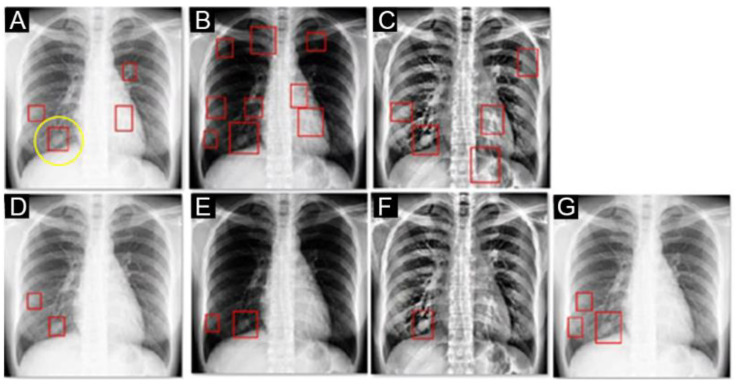
Example of the same CXR with different preprocessing and nodules detected with different models (red box). The true lesion is presented with a yellow circle in (**A**). (**A**,**D**) The original CXRs were preprocessed with default normalization, and lesions were detected with the YOLOv4-based default model with score thresholds of 0.0001 and 0.01, respectively. (**B**,**E**) The CXRs were preprocessed with gamma correction and lesions were detected with the YOLOv4-based gamma correction models with score thresholds of 0.0001 and 0.01, respectively. (**C**,**F**) The CXRs were preprocessed with CLAHE, and lesions were detected with the YOLOv4-based CLAHE models with score thresholds of 0.0001 and 0.01, respectively. (**G**) The CXRs were preprocessed with default normalization, and the lesions were detected with ensemble model, which was derived from the default model at the 0.01 threshold, the gamma correction model at the 0.01 threshold, and the CLAHE model at the 0.01 threshold.

**Figure 9 biomedicines-10-02839-f009:**
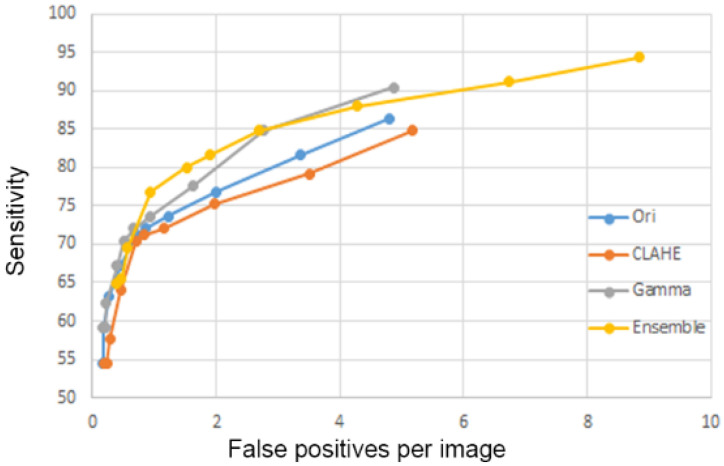
Sensitivity at different numbers of false positives of the original model, gamma correction model, CLAHE model, and ensemble models. CLAHE—CLAHE model; Gamma—Gamma correction model; Ori—original model; FP—false positives.

**Figure 10 biomedicines-10-02839-f010:**
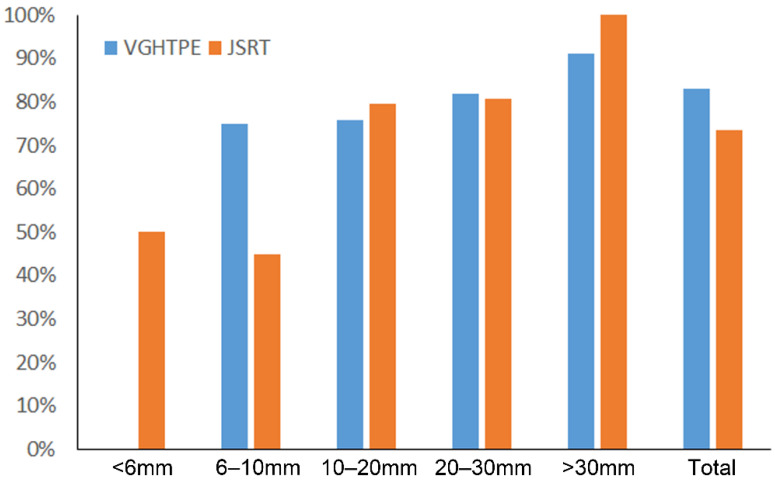
Sensitivity of the AI model (ensemble model) to detect a single lung nodule according to nodule size in the VGHTPE testing dataset and the JSRT dataset.

**Figure 11 biomedicines-10-02839-f011:**
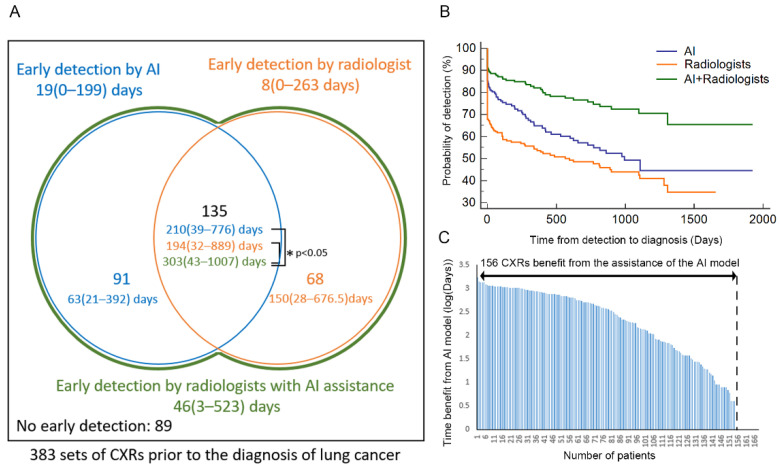
Performance of the radiologists, the trained AI model, and the radiologists assisted with the AI model during retrospective review of CXRs. (**A**) Comparison between the AI model (blue), radiologists (yellow), and radiologists assisted with the AI model (green). (**B**) Time from detection to diagnosis for the AI program, radiologists, and combination of the AI program and the radiologists to simulate the radiologists’ performance assisted by the AI program. (**C**) Waterfall plot of the benefit in the temporal scale of AI assistance in the retrospective review test. AI—artificial intelligence; CXR—chest X-ray.

**Table 2 biomedicines-10-02839-t002:** Characteristics of the VGHTPE training dataset and testing dataset.

	Lung Nodule
N	659
Age, median (IQR)	65 (56–72)
Gender, male (%)	343 (52.0)
Nodule size, N (%)	
<6 mm	34 (5.2)
6–10 mm	44 (6.7)
10–20 mm	260 (39.5)
20–30 mm	163 (24.7)
>30 mm	158 (24.0)
Etiology, N (%)	
Malignant	471 (71.5)
Benign	188 (28.5)

IQR—interquartile range; VGHTPE—Taipei Veterans General Hospital.

**Table 3 biomedicines-10-02839-t003:** Characteristics of the VGHTPE testing dataset and JSRT dataset.

	VGHTPE Testing	JSRT
N	100	154
Age, median	64.5	60.0
Gender, male (%)	56 (56)	68 (44.2)
Nodule size, N (%)		
<6 mm	0 (0.0)	2 (1.3)
6–10 mm	4 (4.0)	29 (18.8)
10–20 mm	29 (29.0)	88 (57.1)
20–30 mm	33 (33.0)	31 (20.1)
>30mm	34 (34.0)	4 (2.6)

IQR—interquartile range; JSRT—Japanese Society of Radiological Technology; VGHTPE—Taipei Veterans General Hospital.

**Table 4 biomedicines-10-02839-t004:** Characteristics of the CXRs in temporal analysis.

	Lung Nodule
N	383
Age, median (IQR)	66 (58–76)
Gender, male (%)	190 (49.6)
Nodule size, N (%)	
<6 mm	0 (0)
6–10 mm	16 (4.2)
10–20 mm	149 (38.9)
20–30 mm	115 (30.0)
>30 mm	103 (26.9)

IQR—interquartile range.

## Data Availability

The datasets generated during and/or analyzed during the current study are available from the corresponding author on reasonable request.

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
