# Peer review of "Artificial Intelligence for Early Detection of Chest Nodules in X-ray Images"

_biomedicines, 2022, doi:10.3390/biomedicines10112839_

Round 1
Reviewer 1 Report
In the Materials and Methods section, the authors should clearly state inclusion and exclusion criteria.
Were all images between 2011 and 2014 performed on the same X-ray machine and protocols?
Author Response
- In the Materials and Methods section, the authors should clearly state inclusion and exclusion criteria.
Response: Thanks for your suggestion. We added the sentence below to clarify this. “The inclusion criteria were as follows: (1) age 20 or over; (2) received CT-guided biopsy or wedge resection; (3) newly diagnosed lung nodule on chest CT. The exclusion criteria were patients with multiple lung nodules on chest CT scans.”
- Were all images between 2011 and 2014 performed on the same X-ray machine and protocols?
Response: No, the CXRs are performed on different X-ray machines and protocols as we stated this problem in “2.3. Preprocessing” and “2.3.1. Image normalization”, and discussed the intermachine variability in ”Discussion” . That’s one of the main reason to do image preprocessing. In the “Preprocessing I” in our study, we standardized the images with monochrome correction and brightness correction.
Reviewer 2 Report
Referee report for Biomedicines - ”Artificial intelligence for early detection of chest nodules
in X - ray images”
The paper exalts the role of artificial intelligence in supporting the clinical practice. It covers in an efficient
way both the preprocessing characterization and the application of algorithms currently available for
nodules detection. The use of an ensemble method is well suited to underline the different perspectives
included in a machine learning procedure. A comparison with radiologist’s detections is described in
detail, with a precise evaluation of temporal resources.
The paper is suitable for publication in Biomedicines. However, the following requirements have to be
fulfilled to improve it.
1. in line 122 the acronym ”CLAHE, contrast limited adaptive histogram equalization” is a refuse;
2. in line 136 the word ”ultilized” has to be replaced by ”utilized”;
3. in line 139 the equation should be centered;
4. in lines 183-185 the datasets used for YOLOv4 should be described more clearly;
5. in caption of Figure 10 you have to explictly refer to the ensemble model;
6. in line 297 you have to specify the type of nodules (e.g. diameter) misclassified by both. In the
following lines 304-311 the correctly classified cases belong to a specific class of nodules?
7. in Figure 11 panel (c) is endowed with a too long x axis, you have to reduce its length, increase
ticks size and explicitly write that it is associated with a quantity (probably number of patients?);
8. in the Discussion, corresponding to lines 338-341, you should include the advantages related with
the detection of metastatic lymph nodes through ensemble methods by including the reference
• A ultrasound-based radiomic approach to predict the nodal status in clinically negative breast
cancer patients, Scientific Reports (2022);
9. in Conclusions you have to add a brief summary about your findings.

Author Response
- in line 122 the acronym ”CLAHE, contrast limited adaptive histogram equalization” is a refuse;
Response: Thanks for your correction. We have deleted the sentence.
- in line 136 the word ”ultilized” has to be replaced by ”utilized”;
Response: Thanks for your correction. We have replaced it with “utilized”.
- in line 139 the equation should be centered;
Response: Thanks for your correction. We have centered it.
- in lines 183-185 the datasets used for YOLOv4 should be described more clearly;
Response: Thanks for your comment. We have added the reference and more description as below:
YOLOv4 achieved 43.5% average precision (AP) and 65.7% AP at an intersection over un-ion of 0.5 (AP50) on the Microsoft Common Objects in Context (MSCOCO) dataset[21], a large dataset containing common objects in our everyday life.
- in caption of Figure 10 you have to explictly refer to the ensemble model;
Response: Thanks for your correction. We have amended “the AI model” to “the AI model (ensemble model)”
- in line 297 you have to specify the type of nodules (e.g. diameter) misclassified by both. In them following lines 304-311 the correctly classified cases belong to a specific class of nodules?
Response: Thanks for your suggestion. We added the relationship between nodule size and early detection or misclassification in Table S1 and Table S2 respectively, and we added below description in line 303-306, “Considering the effect of nodule size, Table S1 listed the probability of early detection according to nodule size. Both the AI model (ensemble model) and the radiologists per-formed better when nodule size increased (Table S2).”.
- in Figure 11 panel (c) is endowed with a too long x axis, you have to reduce its length, increase ticks size and explicitly write that it is associated with a quantity (probably number of patients?);
Response: Thanks for your suggestion. We have amended the figure, including reducing the length of x axis and adding the labels.
- in the Discussion, corresponding to lines 338-341, you should include the advantages related with the detection of metastatic lymph nodes through ensemble methods by including the reference
- A ultrasound-based radiomic approach to predict the nodal status in clinically negative breast cancer patients, Scientific Reports (2022);
Response: Thanks for your suggestion. We have added the reference as below:
“Third, the region of interest (ROI), so called-as the lung segmentation part might reduce computing power and reduce FPs because it excludes extrapulmonary areas. Different preprocessing steps have been used for AI models that detect COVID-19 [22,23]. An ensemble method of different preprocessing methods has been proposed for tuberculosis detection [24]. A study has investigated different ROIs affecting the predictability of nodule status based on ultrasound-based radiomic features in breast cancer patients, and it showed advantage to use ensemble model to enhance the predictive power of the AI models [25].”
- in Conclusions you have to add a brief summary about your findings.
Response: Thanks for your suggestion. We have amended it as below:
“The ensemble model of different preprocessing methods can improve the performance of the AI model. Radiologists may perform better with the assist of AI models in temporal scale. Our results indicate our AI method can aid in the early diagnosis of lung cancer in routine clinical practice.”